# Supercritical Technology as an Efficient Alternative to Cold Pressing for Avocado Oil: A Comparative Approach

**DOI:** 10.3390/foods13152424

**Published:** 2024-07-31

**Authors:** Kelly Roberta Pinheiro Pantoja, Giselle Cristine Melo Aires, Clara Prestes Ferreira, Matheus da Costa de Lima, Eduardo Gama Ortiz Menezes, Raul Nunes de Carvalho Junior

**Affiliations:** 1Program of Post-Graduation in Natural Resources Engineering in the Amazon (PRODERNA), Federal University of Pará, 01 Augusto Corrêa Street, Belém 66075110, PA, Brazil; kelly.sousa@icen.ufpa.br; 2Program of Post-Graduation in Food Science and Technology (PPGCTA), Federal University of Pará, 01 Augusto Corrêa Street, Belém 66075110, PA, Brazil; gisellecsmelo@itec.ufpa.br; 3Food Science and Technology Laboratory (LCTEA), Federal University of Pará, 01 Augusto Corrêa Street, Belém 66075110, PA, Brazil; clara.prestes.ferreira@itec.ufpa.br (C.P.F.); matheus.costa.lima@itec.ufpa.br (M.d.C.d.L.); 4Department of Chemical Engineering, Federal Institute of Education, Science and Technology of Rondônia (IFRO), 4985 Calama Avenue, Porto Velho 76820441, RO, Brazil; eduardo.ortiz@ifro.edu.br; 5Program of Post-Graduation in Food Science and Technology, Program of Post-Graduation in Natural Resources Engineering in the Amazon, Federal University of Pará, 01 Augusto Corrêa Street, Belém 66075110, PA, Brazil

**Keywords:** avocado, supercritical oil, green technology, bioactive compounds, mechanical pressing, supercritical extraction

## Abstract

Avocado oil is rich in nutrients beneficial to human health, such as monounsaturated fatty acids, phenolic compounds, tocopherol, and carotenoids, with numerous possibilities for application in industry. This review explores, through a comparative approach, the effectiveness of the supercritical oil extraction process as an alternative to the conventional cold-pressing method, evaluating the differences in the extraction process steps through the effect of temperature and operating pressure on bioactive quality and oil yield. The results reveal that supercritical avocado oil has a yield like that of mechanical cold pressing and superior functional and bioactive quality, especially in relation to α-tocopherol and carotenoids. For better use and efficiency of the supercritical technology, the maturation stage, moisture content, fruit variety, and collection period stand out as essential factors to be observed during pre-treatment, as they directly impact oil yield and nutrient concentration. In addition, the use of supercritical technology enables the full use of the fruit, significantly reducing waste, and adds value to the agro-industrial residues of the process. It produces an edible oil free of impurities, microorganisms, and organic solvents. It is a green, environmentally friendly technology with long-term environmental and economic advantages and an interesting alternative in the avocado market.

## 1. Introduction

Avocado is a creamy, fatty fruit that originates from the avocado tree (*Persea americana* Mill.) in Central America, mainly in Mexico. Due to its pleasant taste and tropical nature, it is widely sought after by consumers, being recognized as a source of health benefits. Mexico leads the world in avocado production, with approximately one-third of global production, followed by Latin American countries such as Peru and Colombia and others such as Indonesia, Kenya, and Israel [1,2]. 

Avocado consists of approximately 65% pulp, 20% seed, and 15% peel. The oil extracted from avocados is a rich source of fatty acids and triglycerides, along with a high content of bioactive compounds. These properties make avocado oil attractive to the consumer market due to its associated benefits for the cardiovascular system and its anti-inflammatory, antioxidant, anti-cancer, and antimicrobial effects [3,4,5,6]. Furthermore, daily consumption of avocado is also linked to protection of the immune system and mitigation of oxidative damage caused by cellular metabolism. These benefits explain the overall increase in avocado consumption, as the fruit is a rich source of health benefits for humans [3,7].

According to the study by Lieu et al. [2], avocados ripen after harvesting due to the fruit’s vigorous respiration, which accelerates biochemical reactions. High rates of ethylene production during this process cause significant changes to the exterior of the fruit, making strict post-harvest control necessary. The high content of unsaturated fats, along with the substantial nutritional value of avocados [3], accelerates their deterioration when proper storage measures are not implemented after harvest [8]. However, this is not the only reason for their rapid deterioration [9]; avocados can also be severely attacked by microorganisms and fungi, which exploit the fruit’s abundant nutritional value to proliferate. 

Avocado oil is a highly rich source of fatty acids and triglycerides, which makes it attractive to the consumer market [4,5]. In addition, daily avocado consumption is also associated with protecting the immune system and mitigating oxidative damage caused by cellular metabolism. These benefits explain the overall increase in avocado consumption (Figure 1), as this fruit is a rich source of benefits for human health [3,7]. The global consumption of the fruit, for instance, was 6 billion pounds in 2000 and, after 20 years, increased to 18 billion pounds. The trend indicates that avocado is likely to become the second most traded fruit in the world by 2030, only behind bananas, with the United States being the largest consumer [10].

The fruit is consumed in salads, pastas, creams, and ice creams using either the pulp or oil. Due to its composition being rich in bioactives such as tocopherols and monounsaturated fatty acids such as oleic acid and phenolic acids, avocado is an antioxidant source, also rich in carotenoids, which help against free radicals, aid in skin health, protect against cardiovascular diseases, and reduce the cardio-metabolic risk [11,12,13,14,15]. Avocado oil can be an excellent substitute for olive oil in food preparation and as an input for cosmetics and pharmaceuticals. In many cultivars of the fruit, the growth and development of the plant as well as the appropriate harvest period are factors that provide an avocado with superior quality in relation to nutritional composition, causing an interesting and attractive flavor and functional quality to the global market [3,15].

In this perspective, the world is seeking advancements in the development of green technologies for the extraction of oils and compounds of interest from natural matrices. The expansion of these technologies is evident, due to their advantages and economic benefits. However, the methods currently used on a large scale for oil extraction often demonstrate low efficiency and require multiple refining processes to produce a sensorially acceptable product. The extraction of avocado oil can involve both physical and chemical methods. Though mechanical pressing generally yields less oil, the quality of the oil depends on factors such as temperature and press efficiency. Chemical extraction uses solvents, mainly hexane, which offers higher yields but requires refining processes to remove residual solvents [16]. In this context, supercritical fluid technology stands out for addressing key challenges inherent in traditional approaches, offering a sustainable alternative that minimizes environmental impacts and health risks compared to other solvents [2,5,16,17,18]. 

Supercritical carbon dioxide has excellent solvent power due to its controllable density and viscosity, which can be adjusted by modifying temperature and pressure. It is non-toxic and non-explosive, making it ideal for extracting non-polar substances such as oils and fats, and typically results in a high extraction yield. Supercritical fluid extraction also addresses several specific challenges, including the extraction of heat-sensitive compounds that would degrade under the high temperatures required by conventional methods. Moreover, this technique preserves the bioactivity and integrity of compounds by altering the thermodynamic properties of the raw material, enhancing the quality and availability of the extracted compounds and achieving higher yields through improved penetration of the solvent. This makes it feasible to recover compounds of interest without degrading the product [2,5,16,17,18,19].

In addition, the by-product (peel, seed, and defatted pulp) from supercritical extraction can be successfully applied to food, cosmetics, and pharmaceutical products, as it is a residue free of toxic solvents. This represents a significant attraction for the industry, expanding the possibilities of use and reducing waste [6,20,21].

Due to the imminent importance of the fruit, many technological processes that combine the use of green and sustainable methods with speed, high performance, and quality of the product obtained are progressively being studied for integration into industries. From this perspective, this work aims to present a review of the avocado oil extraction process using supercritical technology as an alternative to the cold-pressing method. This includes a comparative approach between the techniques, evaluating differences in the extraction process stages and examining the effects of temperature and operational pressure on the quality of bioactives and oil yield.

## 2. Avocado Pulp Processing

In this section, we will describe the pre-treatment steps for avocado pulp performed to control the moisture content on agricultural properties, thereby reducing oxidative and microbial degradation. Freezing is the most commonly used method for this purpose. In industrial settings, oven drying and freeze drying are additional options for moisture control through dehydration of the pulp. Unfortunately, information on industrial drying methods for avocado pulp is scarce. To address this gap, several studies have been selected that examine the influence of different drying methods on oil extraction at a laboratory scale.

### 2.1. Stages of Agricultural Pre-Treatment 

This pre-treatment stage can be integrated into the pulp processing for oil production either within the industry itself or conducted at a partnering agricultural property that supplies the avocado pulp (Figure 2), with differences in the method adopted for moisture control. Avocados are fruits with high moisture content, containing approximately 70% water. Consequently, drying or freezing steps are essential to ensure that the yield and quality of the oil are not drastically affected. Pre-treatment serves to increase yield and enhance the bioactive content of the oil [6,22,23,24].

Initially, avocados need proper treatment after harvesting to minimize the effects of pulp oxidation, which can cause loss of color and rancidity. Generally, the fruits undergo stages of collection, washing, and selection [25]. They are collected in plantations and selected based on physical characteristics and the stage of maturation. Mostert et al. [26] found in their evaluation of the chemical profile of avocados at different stages of maturation that ripe fruits have a higher lipid content, which consequently can yield more oil during pulp extraction.

After the collection and selection stage, the avocado must be pulped. Pulping can occur on site at the industry with specialized machinery, or the fruit can be pulped before being sold, right on the farm, and then vacuum-packed to minimize degradation by contact with oxygen and preserve its organoleptic characteristics. The packaged pulps are preferably stored in freezers at −18 °C [22].

The avocado production chain is not yet considered sustainable. During the process, the waste generated (including peels and seeds) constitutes approximately 30% of the fruit. This waste is generally not reused in any subsequent processes and is typically discarded, contributing to significant environmental issues related to waste generation [21,22].

Many studies, mainly concentrated in academia, demonstrate that these residues are powerful sources of bioactives, rich in fiber (4–7%), carbohydrates (23–36%), proteins (1–3%), minerals (1–6%), lipids (2–15%), and polyphenols (flavonoids, terpenoids, alkaloids, acetogenins, phytosterols) [3,18]. These by-products can be utilized in various sectors, such as energy production, functional food ingredients, moisturizing creams, and pharmaceuticals (Figure 3). In recent years, there has been a trend in research focusing on the valorization of agro-industrial waste, which in many processes is now considered a by-product of the extractive chain [27,28,29].

As noted, avocado by-products are important alternative sources of food and inputs that can be applied in various industrial sectors. As a source of food, avocado can be an interesting alternative to meet the population’s food needs, considering that in the coming decades, with population growth, lack of food will become a global problem. By-products from the avocado chain can be used as inputs in the bakery industry, providing nutritional and functional value to bread and being used in the production of cookies and cakes [30]. In the beverage industry, avocado by-products can be used in the formulation of teas and juices, increasing the bioactive potential of these drinks. The use of this fruit’s biomass as an energy source is an alternative for the production of bioethanol, being considered a renewable and biodegradable energy matrix [28,29,30,31,32,33].

In addition, the valorization of these by-products has a strong economic impact and can contribute to a closer look at the disposal chain of these wastes during the pre-treatment phases. The implementation of biorefineries [34,35,36], for example, is a strategy for valuing by-products through a sustainable and profitable chain. The systems adopted in biorefineries seek to include sustainable production mechanisms for the recovery of by-products. Waste recovery strategies can be associated with green technologies, such as vacuum microwave-assisted aqueous extraction (VMAAE), ultrasound-assisted extraction (UAE), pressurized liquid extraction (PLE), and supercritical extraction (SFE), which add value to these materials [4,5,6,12]. 

The potential use of biorefineries for the valorization of by-products must be evaluated for feasibility, due to the individual pre-treatment required for each residue and the challenges in standardizing processing. To make the process viable and flexible, integrating raw materials can increase yield and product quality and serve as an economical tool in waste management. However, the use of efficient technologies for extracting target compounds is necessary. A major economic challenge is scaling laboratory results to large-scale production. Factors like plant design, transportation logistics, and investment costs can be limiting issues, as they may not be attractive to investors and governments. Additionally, biorefineries also generate waste that must be managed sustainably [37].

In a study by Restrepo Serna et al. [38], the authors assessed the economic viability of isolated supercritical fluid extraction processes (which do not utilize residues) and integrated processes (which process residues in subsequent stages for ethanol production and cogeneration) using avocado seeds and peels. The integrated processes achieved profit margins of 47.41% and 43.05%, respectively, compared to 21.40% and 21.14% in isolated processes. The authors suggest that the production cost is offset by the increased profit margins in the integrated process.

### 2.2. Influence of Drying Method on Avocado Oil Yield

One of the pre-treatment steps involves controlling the water content within the fruit. As a method to preserve the quality, flavor, and texture of the pulp, freezing procedures are commonly adopted. However, due to the high moisture content of the pulp, the extraction yield is diminished, necessitating sample preparation steps before extraction. This is primarily because the fat and acid content in the oil is higher when the pulp has reduced moisture [23,26,39]. Additionally, factors such as variety, location, harvest time, and fruit maturity directly influence the quality of both the oil and the fruits.

To extract the oil, it is necessary to remove the peel from the fruit and apply treatments to the pulp to reduce its moisture content. When exposed to air, the fruit pulp becomes more susceptible to enzymatic oxidation and the proliferation of microorganisms, as the moist pulp provides an environment conducive to their growth [7,16]. 

In a study conducted by Dos Santos et al. [40], the authors evaluated the influence of oven-drying (40 and 70 °C) and freeze-drying methods on the content of bioactive compounds in avocado oil. The research found that by using freeze drying as a pre-treatment, higher concentrations of bioactives such as α-tocopherol, squalene, cycloartenol acetate, β-sitosterol, campesterol, and stigmasterol were obtained in the oil. In addition, the yield ranged from 33 to 56% for oils obtained of freeze-dried pulp by two different extraction methods: mechanical pressing and Soxhlet, respectively. The concentrations of these compounds were significantly affected with the increase in temperature. The authors observed a loss in the nutritional quality of the oil with an increase in the drying temperature in the oven from 40 to 70 °C. In the extraction by Soxhlet compared to cold pressing, there is a reduction in the content of α-tocopherol. The oil yield was higher in pulp dehydrated at 70 °C extracted by Soxhlet (45.20%) compared to mechanical pressing (29.02%). In the research conducted by Krumreich et al. [41], the authors observed that in avocado oil obtained by mechanical pressing from pulp dried in an oven, increasing the drying temperature from 40 to 60 °C not only increased the carotenoid content but also the concentration of phenolic compounds and antioxidant activity.

The influence of oven-drying and freeze-drying methods on the oil yield obtained by supercritical CO_2_ and Soxhlet extraction using hexane was also investigated by Mostert et al. [26]. The highest oil yield was obtained in the Soxhlet extraction (5.5%), using avocado pulps dehydrated by freeze drying then by supercritical CO_2_ (3.1%). 

In another study by Alkaltham et al. [35], the parts of the ripe avocado, including the pulp, seed and peel, were subjected to three types of drying: in the air for 5 days, in an oven at 60 °C for 16 h, and in a microwave for 15 min at 540 W. The oils extracted from the dehydrated parts of the fruit (pulp, seed, and peel) were analyzed in relation to oil yield, antioxidant activity, and phenolic compound content. The highest yield was obtained in the pulp (77.16%) dehydrated with air drying, and the lowest yield was observed in the pulp dried in a microwave (72.44%). The pulp dehydrated in microwaves showed higher antioxidant activity (82.30%) and lower antioxidant activity (40.96%) when dehydrated in air. The content of phenolic compounds was higher in oven-dried pulp (309.48 mg GAE/100 g) and lower when air-dried (88.71 mg GAE/100 g). For the fruit peel, the highest yield was obtained by microwave drying (21.35%), and the lowest by air drying (11.51%). The antioxidant activity was highest in oven-dried peels (83.42%) and lowest in dried peels in the microwave (73.18%). The content of phenolic compounds in air-dried peels was higher (249.80 mg GAE/100 g), and it was lower in microwave-dried peels (187.53 mg GAE/100 g). Finally, in seeds, the highest yield (2.11%) and antioxidant activity (82.63%) were obtained in oven-dried seeds, and the lowest yield and antioxidant activity were observed in microwave-dried (1.67%) and air-dried seeds (77.63%), respectively. The content of phenolic compounds was highest in air-dried seeds (706.97 mg GAE/100 g) and lowest in oven-dried seeds (210.62 mg GAE/100 g).

In the research conducted by Mostert et al. [26], it was observed that freeze drying enhances oil yield in supercritical extraction, and it does not significantly influence extraction using hexane. In this context, Krumreich et al. [42] note that oven drying at temperatures above 40 °C produces oil of better quality and preservation state, as temperatures below this can lead to incomplete breakdown of lipase enzymes, which can cause oil deterioration.

Regarding avocado, depending on the research or industry objectives, vacuum oven drying at 60 °C may be more suitable for extraction by mechanical pressing, as it can improve yield and preserve oil quality [5,6,26,41,42]. On the other hand, in supercritical extraction, the technique of freeze drying is a cold drying method that can occur at low temperatures, which is a useful characteristic for preserving bioactive compounds like those in avocado, including natural antioxidants of interest to the food industry and for use as inputs in the formulation of new biotechnological products [43,44].

Freeze drying is a promising method, because it removes more than 90% of the moisture from the material; a high moisture content in the pulp interferes with oil extraction. In supercritical extraction, for example, water present in the matrix acts as a cosolvent in the process, interacting with carbon dioxide and reducing its ability to solvate nonpolar compounds, which can lead to low yield [4,20,21,45]. In practical terms, the freeze-drying method differs from other drying processes in terms of the water removal process, the amount of water removed, the extension of the product’s shelf life, and the preservation of sensitive bioactives such as vitamins and physicochemical properties [43,44].

Regarding bioactives, research [40,46] indicates that α-tocopherol acts as an antioxidant during oxidative stress, a condition in which oxidation of unsaturated triacylglycerols occurs. This oxidation is caused by exposure to oxygen, incomplete enzymatic inactivation, and high temperatures during the oven-drying process. This may explain the decrease in this compound in both the pulp and the oil. However, phenolic compounds seem to behave differently. The thermal processing—regardless of the presence or absence of oxygen in the pre-treatment stages—can lead to the hydrolysis of glycosidic bonds and esters. This results in an increase in phenolic acids within the matrix, subsequently increasing their content [47,48,49].

From this perspective, the influence of the temperature and drying method on avocado oil quality and process yield is evident. In addition, factors such as the extraction method, extraction time, type of solvent, conservation status of the fruit, and part of the fruit studied should be considered [39,41,42,47]. Moreover, the challenges faced in food drying include difficulties in standardizing the final moisture content, high energy consumption, and the use of high temperatures, which may compromise the nutritional and functional quality of the product. The scalability of the drying process is a significant gap that needs addressing, as the costs associated with implementing and maintaining machinery and the energy demands of methods like freeze drying and microwave drying can be substantial. Additionally, there is a need to integrate new drying methods into the industry without negatively impacting large-scale production. In this regard, the application of artificial intelligence to optimize industrial processes and the use of solar energy or other renewable sources are tools that could help enhance the sustainability of these methods. These technologies aim to reduce production costs, lower CO_2_ emissions, and increase product yield without compromising the functional quality of the products [43,44,50].

## 3. Avocado Oil Production Process in the Industry 

The most easily accessible lipids, because they are structurally simple, are immediately extracted. The most complex lipids are interconnected with lipid membrane polysaccharides. The avocado mesocarp is composed of parenchymal cells and idioblasts, which are cellulose-covered plant cells containing lipid bodies inside. The lipids present in the plant structures of avocados are easily extracted when ripe, as the cell wall of the fruit has less resistance to breakage. In this case, the extraction process must be so efficient as to rupture this membrane and penetrate it, reaching the oily sacs and emulsions located inside these cells. Currently, avocado oil is mainly extracted by the method of mechanical extraction by pressing [5,7,20,23,51,52].

### Mechanical Press Extraction 

Mechanical extraction is one of the most common procedures for obtaining avocado oil in the industry and is considered a green extraction method, dispensing with the use of organic solvents, with yields equal to or slightly higher than 50% and low production cost. The extraction process occurs by compressing the pulp by mechanical or hydraulic pressing at a temperature below 50 °C [53,54]. 

The procedure begins with pulping the fruit, removing the peel and seed in a destoner machine (Figure 4). The pulp is sent to a crusher machine to homogenize it and form an avocado paste. The avocado paste is then directed to a kneading machine for oil extraction. This extraction takes place with the use of hot water and a carefully controlled temperature. In this process, the paste is subjected to a heat treatment process. This treatment aims to improve and facilitate the extraction of the oil [53,55,56]. 

The kneading step is decisive for obtaining high yields and avoiding oil reabsorption by the pulp. Treatments such as the use of ultrasound, exogenous enzymes, control of the amount of heated water, or the addition of the water–enzyme mixture in appropriate proportions can decrease the extraction time, reducing energy costs in the process without affecting the quality of the oil and increasing the process yield. It is important to note that obtaining avocado oil by mechanical pressing requires that the extraction time (kneading step) be sufficient for the oily emulsions to be broken down [1,5,51,52,54,57].

After the oil is extracted, the partially defatted pulp is separated from the oil by a decanter by the solid-liquid system. The separation takes place in a decanter centrifuge, which rotates continuously with the mixture (paste and oil). The equipment is fed with water heated at the same temperature as the mixture to improve the separation of the oil from the paste and other residues such as the peel and seed. At the end of the process, the three phases composed of oil, water, and waste are separated due to the difference in density [52,54,57,58,59]. 

Generally, purification steps are required after extraction. After separation, the oil phase and the liquid phase still have high water and oil content, respectively. Refining steps are conducted to reduce humidity. The oil is centrifuged again at 50 °C to be separated from the water, until moisture close to 0.2% is obtained. Similarly, a considerable amount of oil is mixed with the wastewater that needs to be centrifuged again [52,54,55,58,60]. 

The purification steps in oil extraction typically involve removing residues through filtration after centrifugation, at which point the oils are considered ready for consumption. However, due to deterioration during the extraction stages—such as heating, prolonged exposure to oxygen, the use of water, and the presence of residues—the extracted oil often exhibits altered physicochemical characteristics. This makes refining steps necessary to reduce acidity and rancidity and to improve color and flavor, thus making the oil edible and market-acceptable [61].

Energy consumption during extraction, particularly in processes involving heating, like the drying and mashing stages, is high and results in significant environmental impact, especially if the energy source is not renewable. The emission of CO_2_ from fossil fuels exacerbates the greenhouse effect. Additionally, the high water demand for washing machinery, its use during the oil extraction stage, and the costs for treating water resources all contribute to environmental degradation. Improper disposal of waste and untreated water also poses risks for disease spread and soil contamination [61,62].

## 4. Extraction by Supercritical Technology

In recent years, concern for environmental issues has led to the development of environmental awareness and responsibility in society. Human activities and the possible consequences arising from the scientific and technological activities of various industrial sectors began to be reassessed. As a result, new sustainable methodologies, techniques, and procedures have been adopted to reduce the impacts caused to the environment [20]. Among these changes, the extraction of plant matrices by green technologies that generate products rich in bioactives, such as supercritical fluid (SFE) extraction, reduce the impact caused by the use of aggressive organic solvents, in addition to promoting the reuse of waste generated from the process [7,19,58].

Fluids in the supercritical state have a density like that of liquids, providing high solvation and solubilization power in certain compounds, in addition to having low viscosity, which contributes to the effective transport and penetration of the supercritical fluid through the pores of the plant matrix. It is observed that temperature and pressure variations influence the transport properties, which can increase the contact of the supercritical solvent with the compounds of interest, influencing the solvation power of the solvent, making the technique selective [20,63,64]. 

Through the ease of operational control of temperature and pressure, supercritical technology also contributes to the conscious and reduced use of solvents, dispensing with the exacerbated use of enormous amounts of toxic organic solvents, in addition to making it possible to obtain plant extracts that are completely free of contamination [27]. In this sense, several solvents (Table 1) can be used in supercritical technology (ethanol, methane, water), the most common being CO_2_ (carbon dioxide), as these have advantages in their application, especially in the food industry [20].

CO_2_ has a relatively mild critical temperature (31 °C) and critical pressure (73 bar), making it advantageous for extracting thermosensitive compounds while preserving the organoleptic characteristics of plant extracts. It is an inert, non-flammable, non-toxic, and non-corrosive gas when in contact with water. Additionally, CO_2_ is cost-effective, highly pure, and does not generate unwanted chemical degradation reactions, nor does it deteriorate the solutes and matrices during the extraction process. Its solvation power is comparable to that of pentane and toluene [64,65,66]. 

Supercritical CO_2_ has the characteristic of having a high density, which makes it unsuitable for solubilizing very polar compounds. In this case, the use of cosolvents is necessary. The most used cosolvents are green solvents, such as ethanol and water. The use of these solvents facilitates the desorption of the solute in the matrix. They are added during the extraction process through a pump that injects them in a constant flow, being solubilized in the fluid. The use of cosolvents in extraction has the advantage of improving the solvation power of polar substances. They can be used in small concentrations up to a maximum of 20% (*v*/*v*) [27,64,66]. 

To ensure product quality, many studies follow standardization in procedures before and after extraction to minimize degradation and contamination of the extracted material. In this context, depending on the raw material, specific protocols must be adopted and adapted to the production characteristics of the industry. For oleaginous raw materials, this includes standardizing the collection period and composition of the fruit, the storage conditions, and pre-treatment processes such as depulping, drying, and refrigeration. Adjustments in the mass of raw material fed into the extraction vessel, duration of the process, and conditions of the process (such as temperature, pressure, CO_2_ inflow and outflow rates, and temperature of the outlet valves, for example) are crucial. Physical-chemical analyses are also necessary to determine parameters such as the peroxide value, acidity, and unsaturation index, for example. Furthermore, more refined analytical techniques like chromatography are used to assess the chemical composition and potential contaminants in the oil. These methods are crucial for ensuring the quality and safety of the oil for various applications. After extraction, the method of collection, handling, and storage of the product are also factors that ensure minimal loss of the bioactive quality [4,61,67,68]. 

The process of extracting this oil by supercritical technology occurs in two stages, extraction and separation. Initially, the avocado is pulped (destoner machine), homogenized (crushing machine), and dehydrated (drying) and then placed in an extraction cell, and the solvent is pressurized and heated to the desired conditions (Figure 5). The pure oil is collected, and its characteristics are evaluated for production and commercialization [26,39,66,69].

The process occurs as follows (Figure 6). CO_2_ in a gaseous state is directed to a pump that, together with an air compressor, conducts the solvent to the extraction cell, in which the avocado pulp is inserted. In the extraction cell, the critical temperature is reached, and the solvent is pressurized. The CO_2_, now in the supercritical state with temperature and pressure above the critical level, penetrates the matrix, solubilizing the solute. 

The supercritical fluid then flows and comes into contact with the raw material for a certain time. This process commonly increases the extraction efficiency and process yield (A) [4,12,19]. In a second moment, the outlet valve is opened, into which the fluid flows at a certain flow rate inside the matrix, extracting the bioactive compounds that are solubilized in the supercritical solvent (B), and the oil is collected and separated after depressurization of the system (C). As a result, an edible, contaminant-free avocado oil is produced, as is a defatted flour that can be used in some other process [4,20,66].

In supercritical extraction, the thin wall of the parenchyma cells present in avocados is easily ruptured, releasing oil droplets that are stored in the cells’ structure. Higher lipid contents make up the idioblasts that are formed by thick walls and require a thermal, enzymatic, or mechanical treatment to be ruptured. In this case, with supercritical technology, the pressure and temperature variables, which are manipulable during the process, can promote the breakdown of the plant cell wall, and the lipids become exposed and are easier to capture by supercritical CO_2_ or another solvent mixture, resulting in yields close to 50% [6,7,20,23].

## 5. Bioactive Composition of Avocado 

The characteristics of the chemical composition of the fruit provide relevant information about the quality of the oil to be obtained and about improvements in the extraction process, since the oxidative stability of the oil is mainly related to the presence of natural antioxidants, such as fatty acids, phenolic compounds, carotenoids, and tocopherols that make up both the pulp and the avocado oil. To ensure the consistency of findings involving bioactive compounds, quality controls should be implemented, such as standardizing extracts to contain specific concentrations, using validated analytical methods to quantify and characterize compounds, and ensuring quality control of raw materials, processing conditions, and proper storage [3,5]. In this sense, some information found in the scientific literature will be briefly described in this topic. 

### 5.1. Fatty Acids

Avocado oil is composed of between 40 and 80% monounsaturated fatty acids, with oleic acid being the main representative of monounsaturated acids, and palmitic acid the main representative of saturated acids. The variations in the percentages found are related to the type of extraction, variety, and locality of the avocado. The oils extracted by supercritical carbon dioxide and LPG ranged from 56 to 60% in the percentage of oleic acid, and from 26 to 29% in the percentage of palmitic acid [12]. 

On the other hand, the oils obtained by different extraction methods, such as Soxhlet and press extraction, ranged from 57 to 59% oleic acid, respectively, and from 19 (mechanical pressing) to 21% palmitic acid (Soxhlet) [41]. The different localities of the same avocado variety can cause significant variations of between 40 and 60% in oleic acid content [11] and from 17 to 24% in relation to palmitic acid composition. The influence of this parameter is very accentuated when the avocado is harvested at different stages of ripeness. As observed by Manaf et al. [13], the oleic acid contents ranged from 22 to 56%, and palmitic acid ranged from 22 to 36% compared to the ripest avocados. Table 2 shows the variations in the contents of the main fatty acids present in avocados. 

As can be observed, linoleic acid may be present in higher concentrations in relation to palmitic acid. One of the main factors for this change in profile may be linked mainly to the stage of maturation of the fruit. Distinct stages can cause variations in linoleic acid content of up to 10% difference, though without causing major changes in the monounsaturated fatty acid profile. The fatty acid profile in avocado oil refers to the importance of this fruit in the diet due to its evident potential for health, mainly due to its similarity to olive oil, and it is also associated with a reduction in cholesterol and in the risk of developing cardiovascular diseases and diabetes [3,71].

In the work by de Marques at al. [73], the authors investigated the effect of avocado oil supplementation during 90 days in obese mice. The group presenting with insulin resistance were supplemented with avocado oil, which improved insulin sensitivity and led to a decrease in hepatic fat accumulation. This effect can be related to the high content of oleic acid. This fatty acid is responsible for many biological effects. A study carried out by Pegoraro at al. [74] investigated the anti-inflammatory effect of oleic acid in a mouse model of irritant contact dermatitis induced by croton oil. Oleic acid reduced acute ear edema, prevented ear swelling after applications of croton oil, and also prevented the rise in pro-inflammatory cytokine IL-1β levels.

### 5.2. Phenolic Compounds 

Phenolic compounds are formed by an aromatic ring with at least one hydroxyl (OH) associated with some functional group. The compound is classified according to the functional group to which it is attached, along with the number of benzene rings and hydroxyl substituents. Avocados are mostly rich in organic acids, which are precursors of fatty acids, flavonoids, and other polyphenols [72,75]. In the fruit, the following acids are identified in higher concentrations: epigallocatechin (1.03 mg GAE/100 g), quercetin (0.557 mg GAE/100 g), caffeic acid glucoside (0.270 mg GAE/100 g), ferulic acid (0.19 mg GAE/100 g), 5-feruloylquinicacid (2.11 mg GAE/100 g), coumaric acid (0.64 mg GAE/100 g), p-coumaric acid (0.58 mg GAE/100 g), p-coumaric acid glucoside isomers (2.62 mg GAE/100 g), p-coumaric acid rutinoside (0.45 mg GAE/100 g), and tyrosol-hexoside-pentoside (0.63 mg GAE/100 g) [3,75]. 

Among these substances, Epigallocatechin, for example, has many biological properties [76]. It harnesses mechanisms that provide protective effects against endometrial, breast, and ovarian cancers in both in vitro and in vivo studies. This compound regulates the activation of nuclear factor erythroid 2-related factor 2, inhibits nuclear factor-kappa B (NF-κB), and can also prevent epithelial–mesenchymal transition. 

Studies carried out on avocado oil indicate the low concentration of phenolic bioactives in this matrix when compared to other parts of the fruit, a fact possibly related to the low level of interaction of these substances in the oil (Table 3).

As observed, extractions from avocado pulp using water or an organic solvent can yield higher levels of phenolic compounds, as demonstrated in maceration or Soxhlet extractions [5,65]. Conversely, depending on the operational conditions, extracting these compounds using carbon dioxide as a supercritical fluid might not effectively solubilize these substances. For instance, operational conditions ranging from 40 to 80 °C under high pressure (400 bar) were found to be ineffective at solubilizing these bioactives [4]. High pressures increase the density of the solvent and its solvation power; however, this also renders the fluid very non-polar, which complicates the solubilization of polar substances [20].

In another study [5], variations in the content of phenolic compounds were observed with two different methods of oil extraction. The content measured was 111.27 mg GAE/100 g using sub-critical carbon dioxide and 130.17 mg GAE/100 g using ultrasound-assisted aqueous extraction. In comparison, the content of these compounds in supercritical oil extraction varied between 33.82 and 50.38 mg GAE/100 g [19], and the content measured using Soxhlet extraction was approximately 33.82 mg GAE/100 g.

The content of phenolic compounds in the oil obtained from extraction by mechanical pressing in research conducted by Krumreich et al. [41] and Krumreich et al. [42] was influenced by the drying method adopted in the pulp pre-treatment stage. Pulps dried in an oven at 60 °C exhibited higher levels of phenolic compounds in the oil of 77.35 and 55.00 mg GAE/100 g, respectively.

The phenolic content varies with the stage of maturation. In a study carried out by Al-Juhaime et al. [23], unripe avocados showed a content of phenolic compounds in the pulp of 106.69 mg GAE/100 g, whereas ripe avocados showed slightly higher results of 113.79 mg GAE/100 g. The fruit variety can also affect the content of these bioactives in the fruit pulp. In research carried out by Golukcu and Ozdemir [77], the Fuerte variety exhibited 0.157 mg GAE/100 g, and the Hass variety 0.246 mg GAE/100 g. 

In another study [78] using the Hass and Shepard types, there was a total phenolic content as a function of catechin (EC) in the peel of 25.32 and 15.61 mg EC/g and in the seed of 9.51 and 13.04 mg EC/g. Velderrain-Rodríguez et al. [79] evaluated the composition of these substances in the peel and seed of the fruit, finding values of 309.95 and 232.36 mM GA/100 g, respectively. 

These findings reveal the importance of the moment of collection in the choice of healthy and ripe fruits to obtain products with excellent bioactive and functional quality, in addition to enabling the exploration of by-products for other possible applications apart from food. In addition to the evident influence of the extraction method and temperature on the chemical composition of avocado extracts, it is also possible to point to the interference of the plant’s cellular metabolism. During growth and development, it can concentrate certain compounds in specific parts of the fruit [65]. 

### 5.3. Carotenoids

Caratenoids are pigmented fat-soluble compounds that are responsible for plant growth and development. They are found in some fungi and bacteria and are considered precursors of vitamin A. They have antioxidant activity, prevent LDL oxidation, reduce the chances of developing coronary heart disease, and are responsible for the greenish, yellow, and reddish colors of vegetables. They are classified according to the presence of oxygen in their structure, and their best-known representatives are lycopene, zeaxanthin, lutein, and β-carotene [3,4,15].

In avocados, lutein is the predominant component [3]. Research conducted on the Hass variety [80] found that ripe fruits have a higher concentration of this compound, ranging between 53.3 and 74.10 µg lutein/g. Avocados harvested across different seasons (January to September) and from various locations (Ventura, San Luis Obispo, San Diego, and Riverside) showed an increase in carotenoid concentration towards the end of the harvest season. Specifically, in Ventura, lutein content in the pulp varied from 3.72 to 8.03 µg lutein/g, and in San Luis Obispo, it ranged from 2.82 to 8.42 µg lutein/g. In the San Diego and Riverside regions, the concentrations were noted to be between 6.63 and 6.29 µg lutein/g and 4.31 and 6.02 µg lutein/g, respectively. This study also identified higher lutein levels in fruits with dark green skin, slightly greenish pulp, and a center near the seed that is slightly yellowish [81].

Variations in carotenoid content can be influenced by the extraction process and the temperature of pulp-drying pre-treatment. As noted by Krumreich et al. [41], using dry pulp at 40 °C resulted in an oil with 71.95 μg β-carotene/g, and increasing the temperature to 60 °C increased the β-carotene content to 88.72 μg/g when using solvent extraction. For mechanical press extraction, the β-carotene content was 75.00 μg/g at 40 °C and 104.62 μg/g at 60 °C. In another study by King-Loeza et al. [65], the composition of carotenoids in the oil ranged from 15.21 to 32.78 μg β-carotene/g. In the peel, β-carotene varied from 13.96 to 15.01 μg/g, and in the fruit pulp, from 6.47 to 9.34 μg/g, indicating that the highest concentration of this compound was found in oil obtained by solvent extraction with enzymatic treatment. 

Although lutein is a major component among the carotenoids in avocados, the consumption of 150 g of the pulp cannot provide the necessary amount (40 mg of lutein per day) for a healthy adult. In this case, supplementation may be a more useful alternative. This compound plays a vital role in human health. It is concentrated in the brain and retina and protects the vision from blue light and oxidative degeneration of the macula, in addition to being a good antioxidant for the skin. Even though it is of paramount importance, unfortunately, the carotenoid content in many studies is still evaluated as a function of β-carotene [3,82].

### 5.4. Tocopherols 

Tocopherols are fat-soluble substances that come in different forms (α−tocopherol, β−tocopherol, γ−tocopherol, δ-tocopherol). They are considered natural antioxidants that extend the shelf life of the oil [1,65]. The Hass variety contains approximately 19.7 mg of α−tocopherol/kg in its pulp. A study carried out by Tan et al. [51] indicates variations in α−tocopherol between 69.2 and 226.7 mg/kg in the oil, and in the Breda variety, α−tocopherol values close to 105 mg/kg in the oil were found.

In another study, Corzinne et al. [83] obtained a defatted avocado extract obtained at 40 °C and 60 °C, with 202 mg α−tocopherol/kg, under a pressure of 200 bar. Some studies indicate α−tocopherol as a compound present in avocado oil, with variations in its content influenced by the different stages of extraction temperature, maturation, variety, and seasonality [3,8]. 

The levels of α-tocopherol in the Fuerte and Hass avocado varieties range from 12.8 to 23.2 mg/kg in the fruit pulp using Soxhlet extraction, as reported by Ge et al. [84] in cultivars from China. Conversely, oil extracted by Santana et al. [85] through expeller pressing exhibited α-tocopherol levels influenced by harvest stage and ripening. Peeled ripe fruits showed higher amounts of α-tocopherol in the oil (74.2 to 120.8 mg/kg) compared to oil from green fruits (44.9 to 104 mg/kg) and were less affected by the type of drying chosen as a pre-treatment.

The levels of tocopherols in avocado oil increase with the ripening of the fruit and gradually decrease over time. With great influence according to the fruit variety and type of extraction, supercritical extraction using carbon dioxide as a solvent has shown the best levels of α-tocopherol in the oil [7,79]. 

## 6. Avocado Oils Obtained by Different Methods

Avocado oil extractions have been performed with different techniques in recent decades. These studies compare the supercritical extraction technique with other conventional extraction methods that can also be adopted as a control mechanism in the process (Table 4). 

In this sense, the avocado oil extraction process was studied and carried out by Corzinne et al. [83] in two stages. In the first stage, through a sequenced extraction, supercritical CO_2_ was used for the extraction of lipophilic substances, and in the second stage, the already defatted sample was extracted through a mixture of CO_2_ and ethanol. The levels of vitamin E were evaluated in the plant extracts collected. In this study, avocado oil was also obtained by Soxhlet as a control mechanism. The research concluded that under high pressure (400 bar), in a system that uses only supercritical carbon dioxide, temperature variations influence the yield obtained. At temperatures of 60 °C and 80 °C, the oil yield varied between 62% and 61%, respectively, when compared to the temperature of 40 °C, with a yield of 60%. In the extraction of Soxhlet, the yield was 65%. In the second stage, it was observed that when organic solvents such as ethanol are used as cosolvents in supercritical extraction processes, the yield can be negatively affected (1.9 and 3%) with the increase in temperature under isobaric conditions (60 °C and 80 °C/400 bar). The extracts obtained using ethanol as a cosolvent showed high concentrations of tocopherols (154 to 282 mg α-tocopherol/kg oil). 

It is important to emphasize that the use of organic solvents as modifiers in supercritical extraction generates a change in the density and polarity of the fluid The flow of CO_2_ promotes the dragging of substances with low polarity, and the ethanol solubilized in the fluid captures the most polar compounds, which increases the bioactive content in the plant extracts obtained, such as the concentration of vitamins in the oil [8,88,89].

Expeller pressing is a single-step pressing method, in which the sample is subjected to high pressure. Compared to cold pressing, the temperature variations are smaller, giving the oil obtained superior quality. Santana et al. [85] evaluated the quality and yield of avocado oils by expeller pressing. In their research, ripe avocados showed high yield (66.7%) compared to unripe avocados (32.7%). However, the concentration of α-tocopherol in the oil of unpeeled green avocados was higher (152 to 210 α-tocopherol/kg oil) than in that of ripe avocados (116 to 120 mg α-tocopherol/kg oil), corroborating the research of Mostert et al. [26], who had already found the strong influence of the stage of ripeness of the fruit on the quality of the oil.

Avocado oil extracted by supercritical CO_2_ compared to that obtained by compressed liquefied petroleum gas (LPG) demonstrates that the extraction method and the type of fluid are parameters that can significantly influence the extraction of antioxidants from the fruit. The extraction by supercritical CO_2_ (40%) showed a lower yield than that observed with LPG (60%); however, the supercritical avocado oils exhibited up to 80% antioxidant activity when compared to the oils obtained by LPG (30%). The use of pressurized fluids such as LPG in relation to the use of supercritical CO_2_ yields a reduced extraction time and uses low temperature and pressure conditions, which reduces energy expenditure. However, CO_2_ is classified as Generally Recognized as Safe (GRAS), presenting different advantages than LPG. It is non-flammable, nontoxic, and safe, making it more advantageous for its use in an increasingly sustainable market [12].

In another study [4], the authors evaluated the bioactive composition of the oil extracted by different temperature and pressure conditions using supercritical CO_2_. The contents of phenolic compounds (15.9–19.1 mg GAE/kg oil), carotenoids (243.6 to 446.8 mg/kg), and oleic acid (40.7 to 69.1%) were found. The oil extracted by cold pressing had a phenolic content of 27.7 mg GAE/kg oil, a carotenoid concentration of 169.2 mg/kg, and an oleic acid content of 60.7%. 

Similarly, in the research conducted by Krumreich et al. [41], the authors obtained mechanically pressed oil with a phenolic compound content ranging from 46.07 to 77.35 mg GAE/100 g. This demonstrates the low solubility of these compounds in carbon dioxide as a solvent, as previously observed by Vilca et al. [4] and Tan and Drang [19]. These studies suggest that oils obtained through mechanical pressing have a higher concentration of phenolic bioactives, whereas supercritical oils have a higher concentration of carotenoids. The results indicate that for higher yields, optimization of the temperature and pressure parameters in supercritical extraction is essential to ensure the preservation of bioactive compounds. The low concentration of carotenoids in mechanically pressed oil is associated with difficulties in breaking down the plant cell wall, which partially extracts the lipid content [7,20].

The contents of fatty acids and phenolic compounds were evaluated in relation to pressure and temperature variations [19] in avocado oil extraction. In the experiment, it was found that fatty acids (0.32%) and total phenolics (416.9 mg GAE/kg) have high concentrations when extracted at 50 °C and 300 bar when compared to lower pressures from 150 to 250 bar. The fatty acids ranged from 0.19 to 0.26%, and the total phenolics from 503.8 to 453.3 mg GAE/kg, respectively. 

In supercritical extraction, the significant influence of pressure and temperature on the solubilization of phytochemical compounds can be observed under constant temperature, where the increase in pressure causes an increase in the extraction yield, which can also have its efficiency optimized with the reduction in moisture in the sample during the pre-treatment stages [20,61,69]. In addition, at high pressures, the density of the fluid increases, improving the solvation power. Because they have more affinity and polarity with certain densities of the solvent, some substances end up being solubilized under different conditions and quantities, which can cause a reduction in the content of bioactives extracted throughout the process [21]. 

Studies such as the one by Botha and McCrindle [86] and Wang et al. [6] demonstrate that the oil extraction time with supercritical technology can be up to five times faster and provide yields similar to those obtained by conventional methods, providing an oil of superior bioactive quality and free of organic solvents.

Also, from this point of view, the variations in the concentration in the content of phenolic compounds are also related to the increase in temperature. In isobaric conditions, this starts to compete with the density of the fluid. In these situations, the temperature will cause the vapor pressure of the solute to be responsible for high yields in operating conditions with low density, reflecting the selectivity of these chemical compounds [25,28]. 

Supercritical technology can achieve yields equal to those obtained by conventional methods. It is also worth noting that in relation to conventional methods, the main difference with this technology is the selectivity of the technique, generating a product with superior quality concentrated in bioactives, in addition to the scalability and sustainability of the process [6,12,86].

In the research carried out by Tran and Dang [19], under 300 bar, the increase in temperature from 34 °C to 50 °C had a significant influence on the extraction of polar compounds. The phenolic content ranged from 338.2 to 416.9 mg GAE/kg oil. In this sense, Table 5 presents an overview of the advantages and disadvantages of each extraction process mentioned in this study. 

In view of this information, some considerations should be made about the current process of extraction by mechanical pressing of avocado pulp. Notably, this extraction method requires many steps, though without promoting the depletion of the raw material. During the process, the avocado pulp is exhaustively homogenized and subjected to heating and the addition of water (up to 20%) to improve the efficiency of the process, which results in higher water and energy expenditure [52,54,90]. 

In addition to being a procedure susceptible to lipid oxidation due to the prolonged extraction performed to improve the yield, which may result in a loss in the quality of the oil, the process sometimes needs auxiliary methods to reduce the extraction time and energy cost. The oil obtained after the extraction stage is not an edible oil and is composed of residues (peel, defatted pulp, and seed) and water, requiring purification steps to reduce impurities, consequently generating more cost for the company. The defatted pulp resulting from the process is usually not reused. Along with the peel and seed, it is discarded, generating a strong environmental impact [52,53,91]. 

The possibility of replacing this extraction technique with supercritical technology has numerous advantages. In principle, the raw material (pulp) is used without the need for any kind of treatment during extraction. Avocado pulp needs to be dried to reduce moisture and be homogenized [23,39], as already discussed in Section 2. After being added to the extraction cell, also called the extraction vessel, the pulp comes into contact with the supercritical fluid (commonly carbon dioxide), with the nonpolar compounds solubilized and extracted in the oil. As supercritical carbon dioxide is a green solvent, it does not leave residues in the collected oil, as it is depressurized and separates from the product [20,21]. 

In addition, the temperature and pressure conditions inhibit microbial growth and decrease the chances of rancidity during extraction, making the product edible soon after the process. Due to the high density of the fluid, it is not necessary to use water in the extraction, which implies less damage to the equipment due to the deterioration caused by the water and less expenses related to water treatment [90]. 

In this sense, it should be noted that the technology adopts mild temperatures, which are sufficient to obtain an oil rich in bioactives, with preservation of thermosensitive compounds. It does not require additional steps to increase the yield, and this can be circumvented by increasing the pressurization of the system at low temperatures. After the pure oil is collected, the defatted pulp (a by-product of the process) is free of organic solvents and has high concentrations of nutrients, and it can be used for other purposes [5,51,53,64,66]. Likewise, the skin and seed of the fruit can be valued using the equipment itself. The residues present in the collection vessel or extraction cell after extraction are free of organic solvents harmful to human health. It should also be noted that depressurized carbon dioxide can be recaptured and reused [6,20,21,92].

It is important to note that although extraction by supercritical technology does not require many steps and provides an excellent quality of oil, the initial cost of implementation can be high, especially for entrepreneurs new to the area. In this sense, special care must be taken, such as a specialized workforce and training of the staff who will work with the equipment under high pressure. In this case, in the cost-benefit analysis, obtaining a pure oil, with high yield and quality, with reuse of waste, free from adjacent steps, through a green process and with high technology, can provide greater market value to this sector. This technology is increasingly sustainable, with better production profitability [20,38,66].

In comparison to conventional extraction technologies, these methods are characterized by the massive use of organic solvents, as well as the energy expenditure required for the evaporation and recovery of these solvents. This is compounded by the difficulty of preserving product quality due to the use of high temperatures and the toxicity of the solvents. Inadequate handling of solvents and the release of toxic gases also present environmental risks, especially in large-scale production [54,92,93].

The environmental impact of supercritical fluid technology should be assessed beyond the significant reduction in the use of organic solvents. This technology brings positive impacts such as minimizing the amount of raw material needed for extraction, using CO_2_ captured from the environment, and reducing pollutant emissions. However, the implementation of this technology also has negative impacts, including high energy consumption and the generation of waste that needs to be properly managed. Additionally, the massive production of CO_2_ for feeding the technology, if obtained from industrial processes like oil refineries, contributes to environmental pollution. Alternatives such as the use of renewable energy sources, waste management, and valorization of these materials through biorefineries, as well as CO_2_ capture and recycling technologies, are possibilities that can minimize the negative impact of the process [94,95,96]. 

In the literature, research focusing on the economic viability of extraction technologies is scarce, yet some studies do evaluate the economic feasibility of mechanical pressing and supercritical fluid extraction techniques. Gonsalves and Teixeira [97] report that the initial cost to establish a mechanical pressing plant with a capacity of 500 kg/h is approximately one million USD. For the implementation of a supercritical extraction unit, research by Almeida [98] on orange peel oil extraction estimates that the equipment cost alone for a capacity of 3 t/h, according to calculations previously made by Prado et al. [99], could exceed two and a half million USD.

In this context, challenges related to the viability of supercritical technology should be mentioned. One of the primary considerations is the adaptation of industrial plants to specific demands and projects. They must meet the unique processing needs and adapt to the characteristics of the raw materials, such as in the valorization of by-products and oil extraction [100]. Regarding production expansion, the studies previously mentioned in this work [4,6,26,87] provide detailed parameters on input and output conditions that can certainly be utilized in scale-up projects. As noted throughout this article, the technology involves a high initial cost for acquiring pre-processing equipment and specific machinery such as pumps, compressors, extraction vessels of appropriate dimensions, and cooling systems. The cost of acquiring raw materials and operational expenses must also be considered [101]. However, despite the significant initial investment, this expenditure is offset by increased production in the industry, with higher market value of the product [101,102]. 

## 7. Trends and Perspectives

Avocado oil and its derivatives represent a broad frontier of interest in food science, nutrition, health, and agriculture, with growing interest in their bioactive properties and versatile applications [59,63], in the development of more efficient and sustainable extraction technologies [84], and in deepening the health benefits associated with their consumption, applications, and functionalities, in order to verticalize the global market [59,65,103]. Sustainability and the optimization of the use of by-products are recurring themes, as is the need for technological advances to overcome production challenges [7,85].

A closer look at trends and perspectives, especially on a topic so rich in applications, can be illustrated by analyzing the predominant scientific aspects in the field. Table 6 is from the Scopus database. Using the descriptors “Persea americana” and “oil” from 2000 to 2023, it elucidates the areas of greatest interest and research related to avocado oil and suggests gaps and opportunities for future studies. These data reveal important insights into key areas of research, as well as the shortcomings and opportunities for studies related to avocado oil. 

It should be noted that the largest proportion of documents is concentrated in agricultural and biological sciences (36.4%). Areas such as chemistry (11.7%) and biochemistry and genetics and molecular biology (11.3%) also contributed to research on avocado oil. The areas of medicine (7.0%) and pharmacology, toxicology, and pharmaceuticals (6.2%) are also of interest for the use of avocado oil. Other fields, such as environmental science, chemical engineering, and engineering, also show considerable interest in avocado oil. However, the lower presence in areas such as energy and nursing indicates the possibility of expanding research in these domains. These data highlight the multidisciplinary nature of avocado oil research and point to the need for interdisciplinary collaborations to fully exploit its potential.

The Hass variety is popular for avocado oil production due to its excellent fruiting characteristics, high yields, and peel properties that protect the fruit during transport [3]. On the other hand, the cold-pressing method and the extraction of supercritical carbon dioxide are “green” techniques, which preserve the functional and sensory properties of the oil, ensuring high nutritional quality [104].

However, when processing avocado oil, various by-products are generated, such as seeds, peels, decanter bagasse, and water. These by-products contain bioactive phytochemicals, whose profile and content are influenced by several factors. These include the climate, irrigation, ripeness, fruit variety, and method of extraction [105]. According to the literature, the seed and bark are rich in phenolic compounds. Phenolic acids predominate in the pulp, whereas procyanidins stand out in the seed and peel [7,106]. These compounds have the potential for use in the food, cosmetic, and pharmaceutical industries, bringing environmental and economic benefits.

Recent studies have emphasized various facets of avocado and its oil, from management and processing techniques to nutritional composition and health benefits. The study in [107] explored techniques to preserve and improve avocado quality and longevity while maintaining its food value throughout the supply chain with strategies such as freezing and microwave drying [14], focusing on the composition and anti-inflammatory and antioxidant activities of avocado oil from different cultivars and emphasizing the chemical variability and its impact on biological properties, such as bactericidal and anti-inflammatory action. In addition, Vilca et al. [4] examined the extraction of oil from the Hass and Fuerte varieties with supercritical carbon dioxide, indicating that the extraction method significantly influences the physicochemical characteristics, fatty acid composition, and antioxidant capacity of the oil, evidencing the need for careful selection of the method to preserve the desired properties [91].

In a broader therapeutic panorama, studies such that by as García-Berumen et al. [108] have identified benefits of this oil in improving mitochondrial function and reducing inflammation and oxidative stress in the context of nonalcoholic fatty liver disease (NAFLD). Studies such as the one by Felemban and Hamouda [109] revealed the ability of avocado seed oil to improve obesity-related parameters in rats. At the same time, Silva et al. [110] indicated that the oil may offer protective effects against benign prostatic hyperplasia. Meanwhile, Wang et al. [6] highlighted the influence of pre-drying methods on oil quality and yield, emphasizing the relevance of the avocado variety used. 

In addition, Shehab et al. [111] successfully explored the use of avocado oil, among other natural products, in the treatment of polycystic ovary syndrome in female rats, evidencing its beneficial effects on the biochemical parameters associated with the condition. These studies point to avocado oil as a substance of interest for interventions related to various health conditions and production strategies.

The applicability and benefits of avocado by-products and properties have been highlighted. There is an example in the study by Corella-Salazar et al. [112], which showed that a phenolic extract of avocado paste can promote satiety in rats, potentially through hormonal mediation. Nasri et al. [103] explored the chemical composition and antioxidant and antibacterial properties of essential oils from avocado varieties, emphasizing their potential in medical and cosmetic applications. 

Cheikhyoussef and Cheikhyoussef [113] discussed the use of avocado waste in various industrial and nutraceutical applications, emphasizing the importance of exploiting these by-products to minimize waste and enhance their economic and industrial uses. Vega-Castro et al. [114] presented avocado seed starch as a viable edible coating to reduce oil absorption and acrylamide formation in potato chips, and Sandoval-Castro et al. [115] explored the genetic diversity of avocado trees in northwestern Mexico, highlighting the need to preserve and exploit this genetic diversity for future breeding programs aimed at adapting to tropical climates.

Another technological trend in the exploration of the avocado fruit is the use of ultrasound-assisted extraction. This method has been gaining prominence in many studies for its ability to increase the yield of oil extracted from the pulp and residues (seed and peel) of the avocado due to the breakdown of the cell wall. When combined with the use of enzymes, it improves the quality, stability, bioactive composition, and shelf life of the oil [58,116]. Studies indicate that this combination increases yield, reduces extraction time, and boosts the content of phenolic compounds [1,51,116].

The combination of technologies has been sparking interest among many researchers. In a study conducted by Barrales et al. [117], the authors combined two extraction techniques (supercritical technology and ultrasound) to achieve an increase in oil yield of up to 8%. Other research by Porto et al. [118] and Liu et al. [119] utilized ultrasound as a pre-treatment before supercritical extraction, in which operational conditions such as temperature and pressure were responsible for the formation of CO_2_ bubbles. These bubbles cause cell wall rupture through cavitation. The integration of these technologies allows for optimization and improved efficiency of processes for forming oily emulsions, which are generated by cellular disruption that is difficult to achieve with conventional methodologies. The challenges in forming these emulsions include inefficient solvent diffusion into cells, varying cellular structures, incomplete cellular disruption, the need for pre-treatment, and the high energy expenditure involved in the process [58,64,120].

In this context, the avocado market should focus on technological innovations in extraction that ensure quality, sustainability, efficiency, and viability of the process. Technologies that minimize the use of solvents, promote the valorization of by-products, and allow for a comprehensive analysis of water consumption, energy, and other parameters for scaling up are crucial. Additionally, there should be a focus on ensuring product quality and preserving bioactive potential through sustainable technologies that enhance bioavailability and industrial applicability. Techniques such as elicitation and metabolic engineering are notable for increasing the content of bioactives in plants and represent significant advancements in the production of natural antioxidants. These methods are particularly valuable, because the low availability of bioactives in some raw materials can deter investment in the industry [96,99,100,117,118,119,120,121]. 

The universe that surrounds avocado oil is rich and multifaceted. The various aspects explored in the current literature, ranging from the optimization of extraction techniques to the exploration of the bioactive properties of this fruit, demonstrate the immense potential still to be unraveled and applied in different sectors. 

## 8. Conclusions

The avocado is a fruit with enormous potential for exploitation, especially in terms of oil extraction and the use of by-products. It has a nutritional composition rich in monounsaturated fatty acids, tocopherol, carotenoids, and phenolics that makes it promising for several areas, from agricultural production improvement to health and industrial applications. The consumption of the fruit is increasing, requiring production processing that is modern, efficient, and sustainable. Supercritical technology emerges as a method of extraction and conservation of target compounds and valorization of agro-industrial residues, avoiding waste and attributing economic and bioactive value to distinct parts of the fruit such as the defatted pulp, peel, and seed. In addition, it is an important alternative to replace the extraction of avocado oil by mechanical pressing, a common method used in industries. When compared, the phytochemical profile of supercritical oil extracts is superior to conventional methods, giving them interesting biological and antioxidant potential beneficial to human health. They are free of organic solvents, residues, microorganisms, and impurities and can be readily applied in various sectors of industry. The optimization of the process parameters and the pre-treatment of the fruit are key factors for the maximum use of the supercritical technology. In this way, encouraging the use of more sophisticated technologies can play a vital role in consolidating avocado oil as an economic, healthy, and sustainable asset, promoting its conscious and innovative use, while supporting scientific and technological advances on a global scale.

## Figures and Tables

**Figure 1 foods-13-02424-f001:**
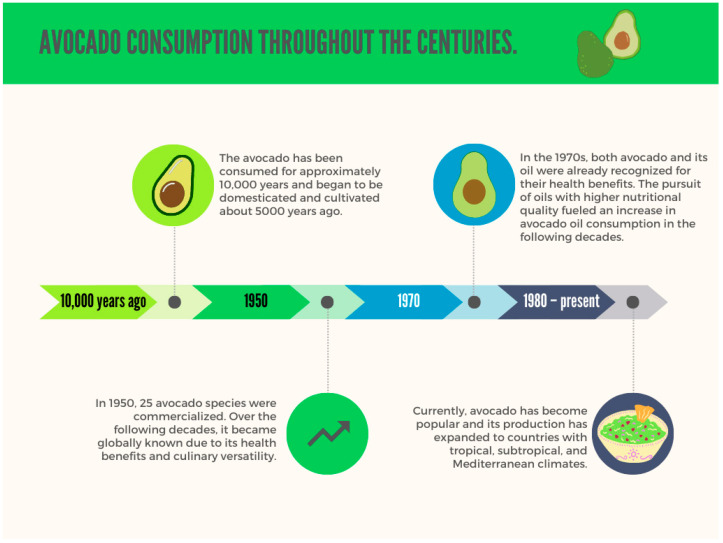
Avocado consumption profile.

**Figure 2 foods-13-02424-f002:**
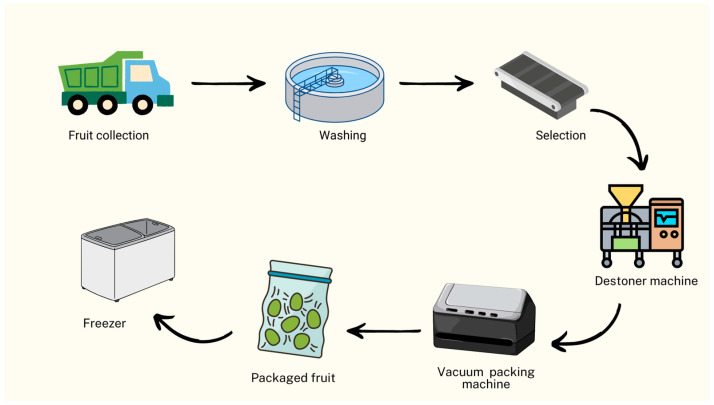
Illustration of avocado pre-treatment procedure after harvest on farms.

**Figure 3 foods-13-02424-f003:**
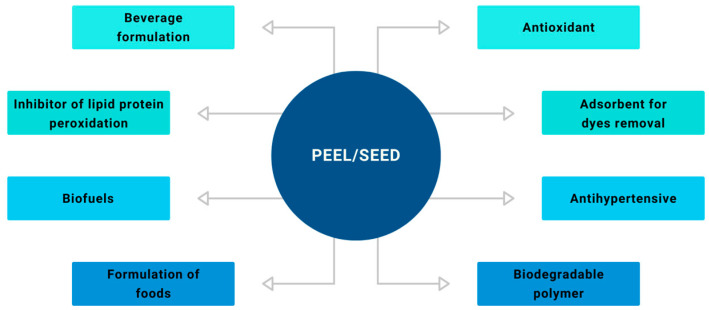
Possible applications of avocado by-products.

**Figure 4 foods-13-02424-f004:**
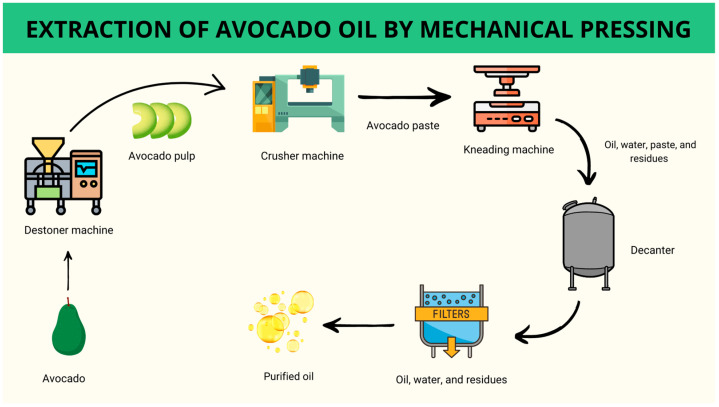
Illustration of obtaining avocado oil by mechanical pressing in industry.

**Figure 5 foods-13-02424-f005:**
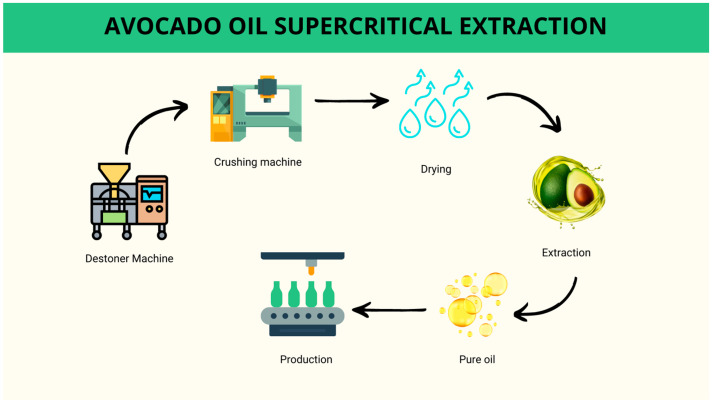
Illustration of obtaining avocado oil by supercritical extraction.

**Figure 6 foods-13-02424-f006:**
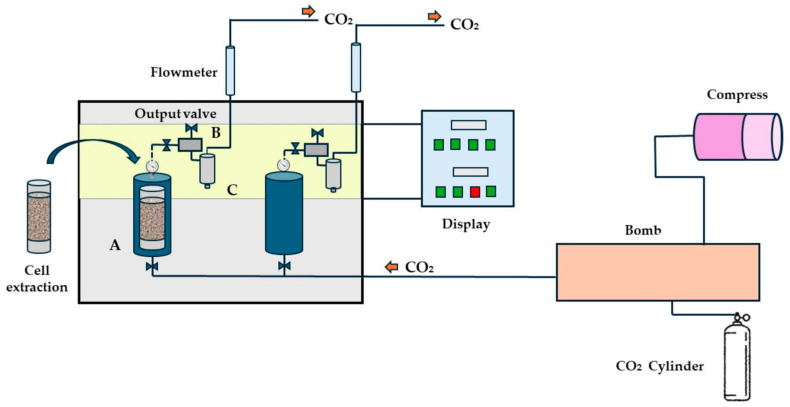
Illustration of avocado oil extraction using supercritical CO_2_ in a pilot plant.

**Table 1 foods-13-02424-t001:** Solvents that can be used in supercritical extraction [20,21].

Solvent	Critical Temperature (°C)	Critical Pressure (Bar)
Propane	16.6	42.5
Water	374	221
Ethane	32.15	49
Ethanol	240.75	61
Methanol	239.45	81
Carbon dioxide	31	73

**Table 2 foods-13-02424-t002:** Major fatty acid profile in avocado oil [11,13,70,71,72].

Monounsaturated Fatty Acid	Content (%)
Oleic acid (C18:1)	22–69
Palmitoleic acid (C16:1)	2.4–19
**Saturated fatty acid**	
Palmitic acid (C16:0)	7–35
Stearic acid (C18:0)	0.5–4
**Polyunsaturated fatty acid**	
Linoleic acid (C18:2)	8.7–55
Linolenic acid (C18:3)	0.1–18
Myristic Acid (C14:0)	0.2–1.6

**Table 3 foods-13-02424-t003:** Phenolic compounds in avocado.

System	Phenolic Compounds in Oil (mg GAE/100 g)	Reference
Maceration	60.5–133.8	[65]
Mechanical pressure	46.07–77.35	[41]
Soxhlet	10.86–39.52	
Mechanical pressure	34.38–55.00	[42]
Soxhlet	15.72–31.21	
Soxhlet	128.17	[7]
Subcritical CO_2_	111.27	
Ultrasound-assisted aqueous extraction	130.17	
Cold pressure extraction	2.77	[3]
SFE	1.09–2.71	
SFE	33.82–50.38	[19]
Soxhlet	33.82	

**Table 4 foods-13-02424-t004:** Avocado oil extraction by supercritical carbon dioxide compared to conventional methods.

System	Solvent	Conditions	Yield (%)	Reference
SFE	CO_2_	40–80 °C/400 bar (120 min)	38.12–39.07	[4]
Cold pressure extraction	-	-	-	
Cold pressure extraction	-	-	25–33	[40]
Soxhlet	Petroleum ether	300 min	45–57	
Expeller pressing	-	-	32.7–66.7	[70]
Cold pressure extraction			50.4–61.2	[85]
Soxhlet	Petroleum ether	-	44.2–55	
Cold pressure extraction	-	-	60	[54]
SFE	CO_2_	40 °C, 60 °C, 80 °C/200 bar, 300 bar, 400 bar (180 min)	12–62	[83]
	CO_2_ + Ethanol (97:3 *v*/*v*)		1.9–31	
Soxhlet	Hexane	-	65	
SFE	CO_2_	34 °C, 42 °C, 50 °C/150 bar, 200 bar, 250 bar, 300 bar (150 min)	30–59	[19]
Soxhlet	Hexane	-	56	
SFE	CO_2_	40 °C, 60 °C, 80 °C/150 bar, 200 bar, 250 bar (150 min)	10–40	[12]
Pressurized fluid	Liquefied petroleum gas (LPG)	20 °C, 30 °C, 40 °C/ 5 bar, 15 bar, 25 bar (10 min)	55–60	
Soxhlet	Hexane	150 min	60	
SFE	CO_2_	37 °C, 81 °C/354 bar, 547 bar (120 min)	56	[86]
Soxhlet	Hexane	480 min	55	
SFE	CO_2_	37 °C/350 bar	67	[26]
SFE	CO_2_	50 °C/400 bar	57	[87]
SFE	CO_2_	50 °C/250 bar (40 min)	56	[6]
Soxhlet	N-hexane	480 min	58	

**Table 5 foods-13-02424-t005:** Overview of advantages and disadvantages of the techniques used for avocado oil extraction [1,2,12,24,40,52,83].

Process	Advantages	Disadvantages
Mechanical press extraction	Green process, high yields, chemical-free, fast process, and low cost	Non-selective, labor-intensive, incomplete extraction, impure oil, higher chance of rancidity
Expeller pressing	Green process, high yields, chemical-free, fast process, and low cost	Oil recovery, residual level, non-selective, labor-intensive, incomplete extraction, higher chance of rancidity
Soxhlet	High throughput, higher extraction efficiency, no manipulation required, longer solute–solvent contact time	Elevated temperature, solvent expenditure, long extraction time, impure residue, toxic solvents, environmental impact, impure oil
Pressurized fluid	Fast process, less volume of organic solvent, high yields, more extraction cycles, green process	Excessive cost of implementation, elevated temperatures, requires pre-treatment of the extraction cell, impure residue
Supercritical extraction	Less volume of organic solvent or water, high yields, green process, higher extraction efficiency, selective, preserves thermolabile compounds, no solvent residue	Hight cost of implementation Hight pressure required

**Table 6 foods-13-02424-t006:** Areas of interest in avocado oil research.

Thematic Area	Nº of Docs	Percentage
Agriculture and life sciences	259	36.4%
Chemistry	83	11.7%
Biochemistry, genetics, and molecular biology	80	11.3%
Medicine	50	7.0%
Pharmacology, toxicology, and pharmaceuticals	44	6.2%
Chemical engineering	35	4.9%
Environmental sciences	34	4.8%
Engineering	26	3.7%
Nursing	26	3.7%
Energy	16	2.3%
Other (immunology and microbiology, materials, multidisciplinary, veterinary, social sciences, physics and astronomy, computer sciences, earth sciences, and economics)	55	8.2%

## Data Availability

No new data were created or analyzed in this study. Data sharing is not applicable to this article.

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
