# Peer review of "Supercritical Technology as an Efficient Alternative to Cold Pressing for Avocado Oil: A Comparative Approach"

_foods, 2024, doi:10.3390/foods13152424_

Round 1
Reviewer 1 Report
Comments and Suggestions for Authors
The authors summarized avocado pulp processing, avocado oil production methods especially supercritical fluid (SFE) extraction method, and bioactive composition of avocado. The review provides interesting information on avocado and its oil processing, which could attract the readers of Foods. However, some issues should be improved.
First of all, according to your text, I sugget the authors change the title, the title did not contain the context of the manuscript.
Line 26, α-Tocopherol, please change to α-tocopherol, and check the whole manuscript.
Line 119, please check the symbol of Celsius degree.
Line 119, 125–126, the symbol - should be changed as – between two numbers , please check the whole manuscript.
Oxford comma should be used, such as a, b, and c, not a, b and c. Please check the whole manuscript.
Line 178–179, there is no space between a number and %.
Line 180, reference 35 “Effect of Drying Process on 784 Oil, Phenolic Composition and Antioxidant Activity of Avocado (Cv. Hass) Fruits Harvested at Two Different Maturity Stages” reported by Alkaltham et al. is out the scope of this review.
Line 182, change minutes to min.
Line 210, the procedures of pressing avocado oil is not consistent with Figure 4. For example, the authors mentioned “removing the peel and seed in a detonation machine” in the text and it did not showed in the figure.
Line 224, change Kneading to kneading.
Line 284, delete “4.1 Supercritical Avocado Oil Extraction”
a three-line table is available.
Author Response
|
Comments 1: The authors summarized avocado pulp processing, avocado oil production methods especially supercritical fluid (SFE) extraction method, and bioactive composition of avocado. The review provides interesting information on avocado and its oil processing, which could attract the readers of Foods. However, some issues should be improved. |
|
Response 1: Thank you for your comment. We agree that the work can be improved. |
|
Comments 2: First of all, according to your text, I suggest the authors change the title, the title did not contain the context of the manuscript. |
|
Response 2: Thank you for the suggestion. The title of the work was revised, lines 2 and 3. Comments 3: Line 26, α-Tocopherol, please change to α-tocopherol, and check the whole manuscript. Response 3: Thanks for the comment. The text was revised, and the term α-Tocopherol was replaced as requested. Comments 4: Line 119, please check the symbol of Celsius degree. Response 4: Thanks for the observation. The text was revised, and the symbol corrected. Comments 5: Line 119, 125–126, the symbol - should be changed as – between two numbers , please check the whole manuscript. Response 5: Thanks for this point. The text was revised, and the symbol corrected Comments 6: Oxford comma should be used, such as a, b, and c, not a, b and c. Please check the whole manuscript. Response 6: Thank you for your considerations. The text was reviewed, and corrections were made. Comments 7: Line 178–179, there is no space between a number and %. Response 7: Thank you for your considerations. The text was revised, and corrections were made throughout the text. Comments 8: Line 180, reference 35 “Effect of Drying Process on 784 Oil, Phenolic Composition and Antioxidant Activity of Avocado (Cv. Hass) Fruits Harvested at Two Different Maturity Stages” reported by Alkaltham et al. is out the scope of this review. Response 8: Thanks for the comment. This observation was pertinent. This reference was added because it deals with the influence of the drying method on different parts of the fruit. The text was improved in lines 234-252. Comments 9: Line 182, change minutes to min. Response 9: Thanks for the comment. The change has been made in lines 236. Comments 10: Line 210, the procedures of pressing avocado oil is not consistent with Figure 4. For example, the authors mentioned “removing the peel and seed in a detonation machine” in the text and it did not showed in the figure. Response 10: We agree with your opinion. Figure 4 has been changed. Comments 11: Line 224, change Kneading to kneading. Response 11: Thank you for this observation. The text was reviewed, and modifications were made. Line 327. |
|
Comments 12: Line 284, delete “4.1 Supercritical Avocado Oil Extraction”, a three-line table is available. Response 12: Thanks for the suggestion. The subtopic “4.1 Supercritical Avocado Oil Extraction” has been removed. |
|
4. Response to Comments on the Quality of English Language |
|
Point 1: |
|
Response 1: Thank you for your observation. |
|
5. Additional clarifications |
|
Not applicable. |
Reviewer 2 Report
Comments and Suggestions for Authors
The manuscript on avocado oil extraction methods provides a comprehensive and critical review of both traditional and advanced techniques, including mechanical pressing and supercritical CO2 extraction. It meticulously analyzes the efficiencies, environmental impacts, and economic feasibility of these methods, offering valuable insights for researchers and industry stakeholders alike. By integrating recent research up to 2023, the manuscript ensures relevance and timeliness in discussing advancements in the field. While the manuscript presents a balanced perspective by outlining both the advantages and limitations of each extraction method, improvements in clarity, seamless citation integration, and inclusion of quantitative data are necessary. However, the manuscript's structured approach effectively facilitates a deeper understanding of avocado oil extraction processes. Overall, with the suggested major revisions to address these points, the manuscript holds promise to significantly contribute to the literature, warranting its acceptance pending the outlined improvements.
Recommendation: Major Revision(s)
Comments to the Authors
1. The abstract needs clearer outlines of methodologies and criteria used for comparing supercritical oil extraction with cold pressing.
2. The introduction should state specific aspects of avocado oil extraction to be compared and evaluated throughout the review.
3. The introduction should delve deeper into specific challenges that supercritical fluid technology aims to overcome.
4. The section on pre-treatment methods lacks clarity on which method is most effective in industry for maintaining oil quality and yield.
5. More justification is needed for why certain pre-treatment methods like freezing are preferred over others in terms of efficiency and bioactive preservation.
6. There should be a clearer synthesis of studies on drying methods to highlight trends, discrepancies, and gaps in current research.
7. Discussions on drying methods should include analysis of their environmental impacts to enrich understanding.
8. More quantitative data on how drying methods affect bioactive compounds like α-Tocopherol and phenolic content is needed.
9. The efficiency of mechanical pressing compared to methods like supercritical CO2 extraction needs specific quantitative comparisons.
10. More detailed discussion on recent advancements in ultrasound and enzyme technologies for extraction efficiency is necessary.
11. The challenges in achieving high separation efficiency, such as emulsion formation, need more detailed exploration.
12. Specific purification methods post-extraction, their effectiveness, and their impact on avocado oil quality require clearer explanation.
13. Discussions on mechanical extraction should include analysis of its environmental impacts, such as energy consumption and waste generation.
14. Specific yield comparisons between supercritical CO2 extraction and traditional methods are necessary for understanding practical advantages.
15. Operational challenges associated with supercritical CO2 extraction, such as maintenance and scalability, should be addressed.
16. Discussions on supercritical CO2 extraction should include a broader analysis of its environmental impacts beyond solvent toxicity.
17. Discussions should include quality assurance measures during supercritical CO2 extraction to ensure oil purity and bioactive preservation.
18. Standardizing units for reporting bioactive compounds across studies would improve clarity and comparability.
19. More analysis is needed on how different extraction methods affect the composition of bioactive compounds in avocado oil.
20. Discussions should include more detailed exploration of bioavailability and clinical evidence linking avocado bioactives to health benefits.
21. Discussions on bioactive compounds should include quality control measures to ensure consistency and reliability of findings.
22. Discussions on avocado bioactives should address environmental and sustainability aspects of production and processing.
23. The high cost and technical complexity of supercritical extraction may limit adoption by small-scale producers.
24. Mechanical pressing's environmental impact, such as waste generation and energy use, should be more thoroughly discussed.
25. Traditional methods' energy-intensive processes contradict sustainability goals due to increased water and energy consumption.
26. Mechanical pressing's potential for incomplete extraction and impurities in oil should be more critically addressed.
27. Discussions on extraction methods should include more detailed quantitative comparisons for sustainability and economic viability.
28. Discussions should include more recent quantitative data and trends in avocado oil research for clarity.
29. Discussions on industrial applications of avocado by-products should include more depth on economic viability and scalability.
30. Discussions should include more detailed exploration of recent innovations in avocado oil extraction technologies.
31. Discussions on health benefits should include more critical evaluation and clinical evidence supporting claims.
32. Discussions on extraction methods should include exploration of emerging technologies for efficiency and environmental impact.
33. The superiority of supercritical extraction should be supported with specific quantitative data and comparative studies.
34. Claims about increasing avocado consumption should be supported with specific data and trends.
35. Discussions should address challenges like high costs and technical requirements associated with adopting supercritical extraction.
36. Discussions should include a comparative environmental impact analysis of supercritical extraction versus traditional methods.
37. Discussions should include a forward-looking perspective on emerging innovations and research directions in avocado oil extraction.
Comments on the Quality of English Language
Author Response
Comments 1: The abstract needs clearer outlines of methodologies and criteria used for comparing supercritical oil extraction with cold pressing.
Response 1: Thank you for your opinion. Lines 24 and 25 have been rewritten for clarity.
Comments 2: The introduction should state specific aspects of avocado oil extraction to be compared and evaluated throughout the review.
Response 2: Thank you for your comment. Done as suggested in lines 89-96.
Comments 3: The introduction should delve deeper into specific challenges that supercritical fluid technology aims to overcome.
Response 3: Thank you for your comment. Done as suggested in lines 97-107.
Comments 4: The section on pre-treatment methods lacks clarity on which method is most effective in industry for maintaining oil quality and yield.
Response 4: Thank you for your comment. Despite the particularities of each drying technique, its advantages and disadvantages, it is also important to consider that the appropriate choice of technique depends on the characteristics of the plant and the objective of the research. In this sense, lines 259-265 were changed.
Comments 5: More justification is needed for why certain pre-treatment methods like freezing are preferred over others in terms of efficiency and bioactive preservation.
Response 5: We agree with your opinion. In fact, freeze-drying is a promising drying technique for the industry due to its characteristics in preserving product quality. Done as requested on lines 266-273.
Comments 6: There should be a clearer synthesis of studies on drying methods to highlight trends, discrepancies, and gaps in current research.
Response 6: Thank you for your opinion. Done as requested on lines 286-291.
Comments 7: Discussions on drying methods should include analysis of their environmental impacts to enrich understanding.
Response 7: We agree with this point of view. Done as requested on lines 292-297.
Comments 8: More quantitative data on how drying methods affect bioactive compounds like α-Tocopherol and phenolic content is needed.
Response 8: Thank you for your comment. We agree that different drying methods influence the bioactive composition of the material differently. New text has been added on lines 224-229, 274-278 and 278-282.
Comments 9: The efficiency of mechanical pressing compared to methods like supercritical CO2 extraction needs specific quantitative comparisons.
Response 9: Thanks for the comment. New text has been added on lines 665-675.
Comments 10: More detailed discussion on recent advancements in ultrasound and enzyme technologies for extraction efficiency is necessary.
Response 10: Thanks for the observation. New text has been added on line 877-890.
Comments 11: The challenges in achieving high separation efficiency, such as emulsion formation, need more detailed exploration.
Response 11: Thanks for the observation. New text has been added on line 890-895.
Comments 12: Specific purification methods post-extraction, their effectiveness, and their impact on avocado oil quality require clearer explanation.
Response 12: Thanks for the suggestion. New text has been inserted on line 341-347.
Comments 13: Discussions on mechanical extraction should include analysis of its environmental impacts, such as energy consumption and waste generation.
Response 13: Thanks for the suggestion. New text insert in lines 348-354.
Comments 14: Specific yield comparisons between supercritical CO2 extraction and traditional methods are necessary for understanding practical advantages.
Response 14: We appreciate your opinion. Table 4 makes a comparison with some avocado oil extraction techniques in relation to yield. To complement this table, a new text was inserted in lines 665-676.
Comments 15: Operational challenges associated with supercritical CO2 extraction, such as -maintenance and scalability, should be addressed.
Response 15: Thanks for the comment. This suggestion was added on lines 785-797.
Comments 16: Discussions on supercritical CO2 extraction should include a broader analysis of its environmental impacts beyond solvent toxicity.
Response 16: Thanks for the suggestion. The new text was added on lines 766-776.
Comments 17: Discussions should include quality assurance measures during supercritical CO2 extraction to ensure oil purity and bioactive preservation.
Response 17: Thanks for the suggestion. The new text was added on lines 396-410.
Comments 18: Standardizing units for reporting bioactive compounds across studies would improve clarity and comparability.
Response 18: Thanks for the suggestion. Done as requested.
Comments 19: More analysis is needed on how different extraction methods affect the composition of bioactive compounds in avocado oil.
Response 19: Thanks for the suggestion. Table 3 has been inserted on page 13 and new text has been added on lines 526-530.
Comments 20: Discussions should include more detailed exploration of bioavailability and clinical evidence linking avocado bioactives to health benefits.
Response 20: Thanks for the suggestion. New text insert in lines 379-487 and 501-505.
Comments 21: Discussions on bioactive compounds should include quality control measures to ensure consistency and reliability of findings.
Response 21: Thanks for the suggestion. New text insert in lines 447-451.
Comments 22: Discussions on avocado bioactives should address environmental and sustainability aspects of production and processing.
Response 22: Thanks for the suggestion. New text insert in lines 896-906.
Comments 23: The high cost and technical complexity of supercritical extraction may limit adoption by small-scale producers.
Response 23: Thank you for your comment. The initial implementation cost is indeed high, but small businesses can also participate through partnerships to acquire raw materials, for example. In addition, the initial investment is amortized by the high market value of the product.
Comments 24: Mechanical pressing's environmental impact, such as waste generation and energy use, should be more thoroughly discussed.
Response 24: Thank you for your feedback. Text has been added to lines 348-354.
Comments 25: Traditional methods' energy-intensive processes contradict sustainability goals due to increased water and energy consumption.
Response 25: Thanks for the comment. We agree with your point of view.
Comments 26: Mechanical pressing's potential for incomplete extraction and impurities in oil should be more critically addressed.
Response 26: Thanks for the comment. The expression “incomplete extraction” may be misunderstood, in this sense, it will be replaced by “partial extraction” or “partially extracted”. In relation to impurities, impurities consist of residues such as remnants of peel, seeds and degreased pulp, which come with oil and need to be separated by centrifugation. Therefore, text was added in line 341-347.
Comments 27: Discussions on extraction methods should include more detailed quantitative comparisons for sustainability and economic viability.
Response 27: Thank you for your comment. New text has been added on lines 348-354, 760-765, 766-776, 777-784.
Comments 28: Discussions should include more recent quantitative data and trends in avocado oil research for clarity.
Response 28: Thank you for your comment. New text has been added on lines 877-906.
Comments 29: Discussions on industrial applications of avocado by-products should include more depth on economic viability and scalability.
Response 29: Thanks for the comment. New text has been inserted on line 186-201.
Comments 30: Discussions should include more detailed exploration of recent innovations in avocado oil extraction technologies.
Response 30: Thank you for your opinion. New text has been inserted in lines 877-883.
Comments 31: Discussions on health benefits should include more critical evaluation and clinical evidence supporting claims.
Response 31: Thanks for the suggestion. This discussion is more detailed in section 5, as the health benefits are directly linked to the bioactive compounds present in avocado oil. Lines 479-488 and 501-506.
Comments 32: Discussions on extraction methods should include exploration of emerging technologies for efficiency and environmental impact.
Response 32: Thank you for the suggestion. We understand your opinion. The work aims to compare supercritical technology, which is an emerging and environmentally friendly technology, with mechanical cold pressing, providing complementary experimental information from other conventional methods. In this sense, this information will be added in the introduction lines: 112-119.
Comments 33: The superiority of supercritical extraction should be supported with specific quantitative data and comparative studies.
Response 33: Thank you for your opinion. We emphasize that the selectivity of the technique generating a product with superior quality concentrated in bioactives, scalability and sustainability are the technology's differentiators. Additional texts have been inserted on pages 665-675, 701-705 and 766-776.
Comments 34: Claims about increasing avocado consumption should be supported with specific data and trends.
Response 34: Thanks for the suggestion. New text has already been added on lines 67-71.
Comments 35: Discussions should address challenges like high costs and technical requirements associated with adopting supercritical extraction.
Response 35: Thanks for the suggestion. New text has already been added on lines 777-797.
Comments 36: Discussions should include a comparative environmental impact analysis of supercritical extraction versus traditional methods.
Response 36: Thanks for the suggestion. New text has already been added on lines 760-765 and 766-776.
Comments 37: Discussions should include a forward-looking perspective on emerging innovations and research directions in avocado oil extraction.
Response 37: Thanks for the suggestion. New text has already been added on lines 896-906.
- Response to Comments on the Quality of English Language
Point 1:
Response 1: The text has been revised.
Reviewer 3 Report
Comments and Suggestions for Authors
Massive proof reading and language correction required
1. Abstract sentence not clear: dispensing with purification steps ?
2. Pls checking intext citations. Numbering incorrect.
3. Very generic introduction. Can be improved.
4. Line 118 Packaged packages are stored in freezers preferably at -18ºC rephrase?
5. Line 139 Pasta is not a bakery product.
6. 149 -microwave-assisted aqueous vacuum extraction (VMAAE) check full form against abbreviation
7. 162 from the fruit, in addition to some treatment on the pulp to reduce the humidity, in the researchers reported in this topic the humidity varied between 3% and 5%, after the dry- ing treatment. Sentence not clear.
8. Massive proof reading and language correction required.
9. Line 170 α- tocopherol and not a-tocopherol
10. Line 183 analysed and not influenced in relation to yield
11. 176-189: please make appropriate distinction between pulp, seed or whole fruit wrt to oil yields and antioxidant activity. Not clear.
12. Line 187 What is air dried activity? -
13. Line 213- low pro-duction cost. Can you give approximate numbers?
14. Line 216- the peel and seed in a deto-nation machine. Is that Detonation or Destoning?
15. A table comparing compositional differences between avocado oil extracted using different techniques would be better than in text comparison.
Comments on the Quality of English Language
no
Author Response
Comments 1: Massive proof reading and language correction required
Response 1: Thank you for your comment. The text has been revised.
Comments 2: Abstract sentence not clear: dispensing with purification steps ?
Response 2: Thank you for your comment. This sentence has been removed.
Comments 3: Pls checking intext citations. Numbering incorrect.
Response 3: Thank you for the observation. Citations have been revised.
Comments 4: Very generic introduction. Can be improved.
Response 4: Thank you for your comment. New text has been added at lines 45-52, 67-71, 89-107.
Comments 5: Line 118 Packaged packages are stored in freezers preferably at -18ºC rephrase?
Response 5: Thank you for the comment. The text has been rewritten for better understanding: The packaged pulps are stored in freezers preferably at -18°C. Line 150.
Comments 6: Line 139 Pasta is not a bakery product.
Response 6: Thank you for the comment, we agree that the sentence should be clarified. The lines 172 and 172 was rewritten to: […] nutritional and functional value to breads, and in the production of cookies and cakes [30] and in the beverage industry […]. Additionally, a new citation was inserted into the text.
Reference:
- Mahawan, M.A.; Tenorio, M.F.A.; Gomez, J.A.; Bronce, R.A. Characterization of Flour from Avocado Seed Kernel. APJMR 2015, 3, 34-40.
Comments 7: 149 -microwave-assisted aqueous vacuum extraction (VMAAE) check full form against abbreviation
Response 7: Thank you for the comment. The sentence was rewritten to: […] vacuum microwave-assisted aqueous extraction (VMAAE) […]. Line 183.
Comments 8: 162 from the fruit, in addition to some treatment on the pulp to reduce the humidity, in the researchers reported in this topic the humidity varied between 3% and 5%, after the drying treatment. Sentence not clear.
Response 8: Thank you for the comment. We agree with your opinion and the authors decided to remove the sentence.
Comments 9: Massive proof reading and language correction required.
Response 9: We agree with your opinion and the text will be revised
Comments 10: Line 170 α- tocopherol and not a-tocopherol
Response 10: Thank you for the comment. The text was revised, and the modifications made.
Comments 11: Line 183 analysed and not influenced in relation to yield
Response 11: Thank you for the comment. The modification has been made. Line 237.
Comments 12: 176-189: please make appropriate distinction between pulp, seed or whole fruit wrt to oil yields and antioxidant activity. Not clear.
Response 12: Thank you for the comment. We agree that the text can be rewritten. Lines 230-252.
Comments 13: Line 187 What is air dried activity?
Response 13: Thanks for the observation. The term “air dried activity” was removed from the text.
Comments 14: Line 213- low pro-duction cost. Can you give approximate numbers?
Response 14: Thank you for the comment. Gonsalves and Teixeira [98] report that the initial cost to establish a mechanical pressing plant with a capacity of 500 kg/h is approximately one million dollars. For the implementation of a supercritical extraction unit, Almeida's research [99] on orange peel oil extraction estimates that the cost of the equipment alone for a capacity of 3 t/h, according to calculations previously made by Prado et al. [100], could exceed two and a half million dollars. The acquisition cost of the supercritical extractor alone for large-scale production is high, but studies suggest that investment capital is amortized with the increase in production, in addition to the higher market value of the product obtained.
Reference
- Gonsalves, R.A.; Teixeira, L.C. Estudo de viabilidade técnica e econômica do processamento do abacate (P. americana Mill.) variedade Hass, visando a extração do óleo. Monography, Federal University of Rio de Janeiro, Rio de Janeiro, 2017.
- Almeida, T.S.S. Estimativa do custo de manufatura da extração com dióxido de carbono supercrítico do óleo de casca de laranja. Monography, Federal University of Uberlândia, Minas Gerais, 2019.
- Prado, J.M.; Dalmolin, I.; Carareto, N.D.D.; Basso, R.C.; Meireles, A.J.A. Oliveira, J.V. Batista, E.A.C. Meireles, M.A.A. Supercritical fluid extraction of grape seed : Process scale-up , extract chemical composition and economic evaluation. Journal of Food Engineering 2012, 109, 249–257. http://dx.doi.org/10.1016/j.jfoodeng.2011.10.007.
Comments 15: Line 216- the peel and seed in a detonation machine. Is that Detonation or Destoning?
Response 15: Thank you for the comment. The text on line 314 has been rewritten [...] removing the peel and seed in a destoner machine […].
Comments 16: A table comparing compositional differences between avocado oil extracted using different techniques would be better than in text comparison.
Response 16: Thank you for the suggestion. Table 3, page 13 has been added. The text has been rewritten.
Reviewer 4 Report
Comments and Suggestions for Authors
The manuscript "Supercritical Technology as an Efficient Alternative to Traditional Avocado Oil Extraction: A Comparative Approach", constitutes an excellent review of the topics of supercritical technology and the possibilities that this technology offers for the extraction of avocado oil.
The work is well structured and written clearly and understandably. We just had to discuss the scalability of the supercritical technology a little bit more and comment a little about the cost of installation and operation of this technology.
Despite the manuscript's undeniable merits, typographical errors were detected in expressing some units (the "°" and "%" must be used separately from the numerical value), and the comma, instead of the dot, has been used to express some numerical values.
These errors must be corrected before publishing the work.

Author Response
Comments 1: Despite the manuscript's undeniable merits, typographical errors were detected in expressing some units (the "°" and "%" must be used separately from the numerical value), and the comma, instead of the dot, has been used to express some numerical values. These errors must be corrected before publishing the work.
Response 1: Thank you for your corrections. The corrections requested in the PDF file were met. However, according to the guidance from the other reviewers, the manuscript must follow the Oxford style, without spacing between the numbers and symbols. The authors also evaluated other publications in the journal for confirmation.